# Gate-Width Optimisation Based on Time-Gated Single Photon Avalanche Diode Receiver for Optical Wireless Communications

Yu Mu , Xiaoxiao Du , Chao Wang, Ziwei Ye and Yijun Zhu *

Communication Department, Information Engineering University, Zhengzhou 450000, China; muyuvlc123@gmail.com (Y.M.); xxgcdxx@163.com (X.D.); xxgcwangchao@163.com (C.W.); yzwleaf@outlook.com (Z.Y.)
* Correspondence: yijunzhu@zzu.edu.cn

**Abstract:** Using a single photon avalanche diode (SPAD) as a receiver in an optical wireless communications (OWC) system can effectively expand the transmission distance. However, the performance of the SPAD receiver is usually affected with the bit error rate (BER) lower bound determined by background light and the inter-symbol interference (ISI) distortion caused by dead time. In this paper, external time-gated technology is employed, and the SPAD is only activated within the set gate-ON time to alleviate the influence of background light and ISI distortion. The SPAD photon counting model and the communication BER model are established according to the external time-gated characteristics. Considering the functional relationship among the gate-ON time, signal light flux, background light flux and blocking time, we take the minimum BER as the optimization target, the optimal gate-ON time is derived. The numerical results show that whenever the signal light flux or the background light flux is higher, the BER performance of the time-gated mode is apparently better than the free-running mode. For example, when signal photons and background photons are 30 and 10 per symbol time, respectively, the BER of free-running SPAD converges to 0.1, while the BER of the time-gated scheme is about $10^{-6}$.

**Keywords:** single photon avalanche diode (SPAD); optical wireless communications (OWC); time-gated SPAD; optimal gate-ON time; background light

## 1. Introduction

The introduction of a single photon avalanche diode (SPAD) in the optical wireless communication (OWC) system can effectively expand communication distance [1,2], and reliable transmission can be established in the case of the limited signal power [3]. However, the lower bound of the SPAD receiver bit error rate (BER) is determined by the background light. The SPAD detector cannot distinguish the background photon and signal photon only in one individual avalanche pulse, which requires a longer symbol time to improve the accuracy of the signal transmission by recording a mass of pulse counts, resulting in a low data rate of the SPAD receiver [4]. In addition, the influence of dead time on SPAD communication performance cannot be ignored. Dead time causes the counting loss and counting block. In a certain symbol period, the dead time after a photon arrival affects the detection of the next photon, resulting in the counting loss. In adjacent symbol periods, dead time may lead to inter slot interference (ISI), resulting in count blocking [5,6]. The symbol time of SPAD is generally longer than the dead time, which makes it difficult to realize high-speed applications. How to use a high-sensitivity SPAD to realize high-speed communication under noisy ambient light conditions has attracted increasingly widespread attention.

In order to realize high-speed communication under the background light, array-receiving pattern can be used for SPAD to expand the dynamic range [7]. Ref. [8] uses a SPAD array with 60 pixels to realize high-speed transmission in 800 lx background light flux

intensity, with the maximum data rate at 200 MB/s. In Ref. [9], the author concluded that for SPAD arrays with the same total effective detection size, the dynamic range is positively correlated with the number of pixels. In order to achieve a high performance detection under the intensive background light condition, the SPAD's array scale has to be increased. This approach may lead to a decrease in SPAD detection efficiency. Ref. [10] proposes that if the photon detection efficiency of SPAD array is higher than 14%, the receiver based on the SPAD array has better sensitivity than the APD receiver in background light. Although large-scale SPAD arrays can achieve faster data rates and larger dynamic range, the effect of decreased detection efficiency on the performance of the receiver cannot be ignored [11].

Although the receiver photon detection efficiency may be affected, the improvement of the dynamic range can provide a stronger ability to resist shot noise. Applying photon detection techniques with large dynamic ranges to the communication field to enhance the anti-background lighting capacity of the receiver is a potential solution, such as the time-gated single photon detection technology [12]. The time-gated SPAD detection scheme can be used to improve the dynamic range of time-correlated single photon counting (TCSPC) [13]. Laser radar and biomedical devices also reduce the photon incidence of specific time through the time-gated scheme to improve the detection accuracy [14].

At present, external time-gated techniques have been widely studied in the photon detection field [15,16], but the use of time-gated schemes in optical wireless communication systems is still in the initial research stage [17,18]. In communication applications, time-handed technology can effectively reduce the effects of background photons. However, time-gated techniques reduce the impact of background photons by decreasing exposure time, but limited exposure times also reduce the incidence of signal photons [19]. Therefore, we build the photon count model and BER model of the time-gated mode based on the photon counting characteristics of the SPAD. We use the minimum BER as the optimization target, give the optimal gate-ON time according to the signal photons, background photons and blocking time to optimize the communication performance of the SPAD receiver.

Our contribution:

- By applying the external time-gated technology to the receiving front end, SPAD is only activated within the set gate-ON time. According to the characteristics as dead time, gate-ON time and blocking time of AQ-SPAD, the photon counting model and BER model are constructed.
- Taking the minimum BER as the optimization objective and the gate-ON time as the independent variable, the optimal gate-ON time is obtained by weighing the signal photons, background photons and blocking time, which is verified by experiments.

According to the research content, this paper is organized as follows: In Section 2, considering the gate-ON time, dead time and ISI distortion, the photon counting model of time-gated AQ-SPAD is constructed; in Section 3, we build the BER model of time-gated SPAD, and improve the receiver performance by optimizing the gate-ON time; the influence of gate-ON time on SPAD performance is analyzed, and the performance of the time-gated scheme and the free-running scheme is compared in Section 4. Conclusions are drawn in Section 5.

## 2. Time-Gated AQ-SPAD Receiver Modeling

SPAD is very sensitive to the arrival of photons. When the photon arrives and triggers the avalanche event, the counting pulse output by SPAD can be detected directly. However, the SPAD cannot respond to incident photons within the dead time after the photon arrival, resulting in counting loss. The dead time of the active quenching single photon avalanche diode (AQ-SPAD) is constant. The photons arrival within the dead time will prolong the dead time of passive quenching single photon avalanche diode (PQ-SPAD). Therefore, the AQ-SPAD has a higher count rate than the PQ-SPAD. The modeling and research of this paper is based on AQ-SPAD.

The difference between the free-running scheme and the time-gated scheme is whether there are additional time-gated signals. For the time-gated SPAD, photon detection only

occurs in the gate-ON time, and the photons cannot be detected at the gate-OFF time. The output of the free-running SPAD and the time-gated SPAD is shown in Figure 1a, the blue dashed portion represents the missed photons. It is worth noting that for AQ-SPAD, regardless of whether the external gating signal is turned on, it cannot detect photons before the end of the dead time. Setting gate-ON time can effectively reduce the error count caused by background photons and alleviate ISI distortion [20].

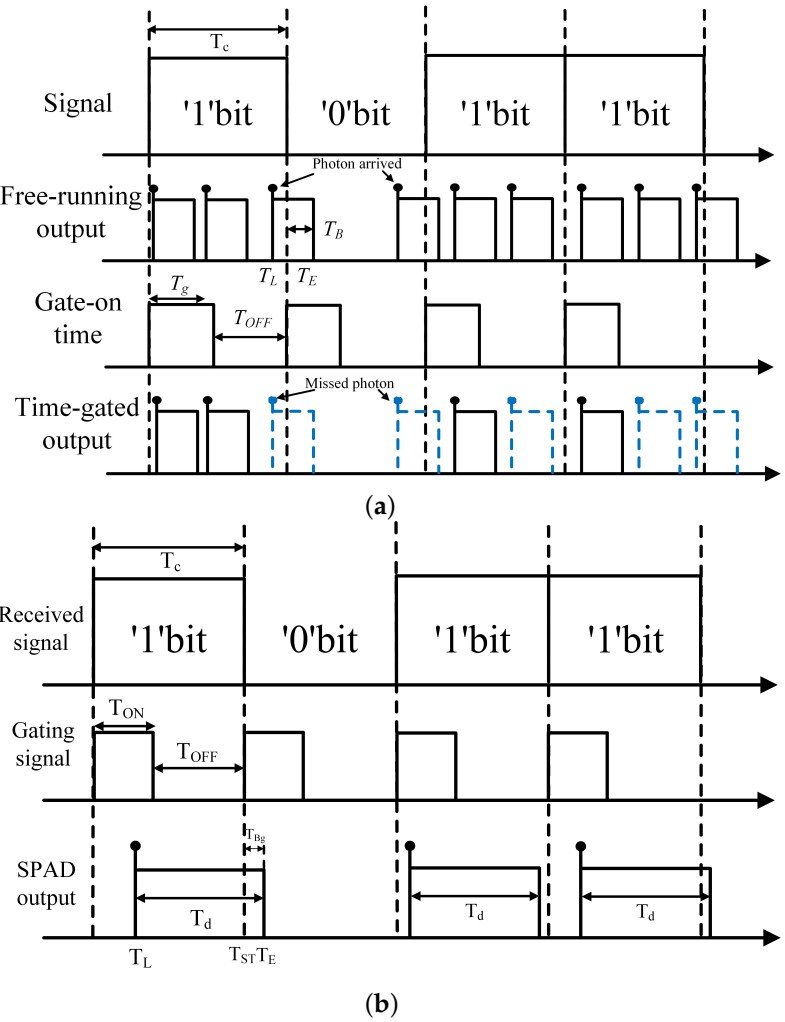

**Figure 1.** The output characteristics of free-running mode and time-gated mode. (**a**) Free-running AQ-SPAD and time-gated AQ-SPAD. (**b**) The output of time-gated SPAD. (If the gate-OFF time is less than dead time, ISI distortion may occur in AQ-SPAD).

### 2.1. Single Time-Gated SPAD Receiver

To describe the performance of time-gated AQ-SPAD, it is necessary to establish a photon counting model to characterize the SPAD. In Figure 1b, $T_L$ is the photon arrival time of the previous symbol. $T_E$ is the end time of the avalanche, $T_{st}$ is the start time of the symbol, and $T_C$ is the symbol duration. Considering ISI distortion, SPAD may have a blocking time after the beginning of the symbol. The blocking time $T_{Bg}$ can be expressed as:

$$T_{Bg} = \max\{T_d - (T_{st} - T_L), 0\} \tag{1}$$

where, $T_d$ is dead time. Considering ISI distortion, the actual detection time of the SPAD is $T_C - T_B$. It is worth noting that the blocking time $T_{Bg}$ is determined by the gate-ON time, dead time, and symbol time.

If the dead time is longer than the gate-OFF time, ISI distortion may occur in time-gated SPAD, and blocking time $T_{Bg}$ meets $T_{Bg} \in [0, T_g]$. If the dead time $T_d$ meets $T_d > T_c + T_g$,

SPAD will not detect photons at the next gated-ON time. Therefore, the relationship between blocking time $T_{Bg}$ and dead time $T_d$ of the gated mode can be expressed as:

$$
T_{Bg} = \begin{cases} 0, & T_C - T_g > T_d \\ [0, T_g] & T_C - T_g < T_d < T_C + T_g \\ T_g, & T_C + T_g < T_d \end{cases}
\tag{2}
$$

For the time-gated scheme, the maximum photon count of the SPAD receiver can be calculated as:

$$
K'_{MAX} = \left\lfloor \frac{(T_g - T_{Bg})}{T_d} \right\rfloor + 1.
\tag{3}
$$

The number of effective incident photons can be calculated:

$$
\lambda_{k'} = \lambda \left( T_g - T_{Bg} - k' T_d \right)
\tag{4}
$$

where, $k'$ represents the actual photons count, $\lambda$ is the photon incidence rate and the unit is photons/s. Then, the probability density distribution function of the time-gated SPAD photon count can be calculated [21,22]:

$$
p_K(k') = \begin{cases} \sum\limits_{i=0}^{k'} u(i, \lambda_{k'}) - \sum\limits_{i=0}^{k'-1} u(i, \lambda_{k'}), & k' < K'_{MAX} \\ 1 - \sum\limits_{i=0}^{k'-1} u(i, \lambda_{k'}), & k' = K'_{MAX} \\ 0. & k' > K'_{MAX} \end{cases}
\tag{5}
$$

where, $u(i, \lambda) = \frac{\lambda^i}{i!} e^{-\lambda}$. Considering ISI distortion, SPAD counting means and counting variance can be calculated [23]:

$$
\mu_K = K'_{\max} - \sum\limits_{k=0}^{K'_{\max}-1} \sum\limits_{i=0}^{k'} u(i, \lambda_{k'}),
\tag{6}
$$

$$
\sigma^2_{k'} = \sum\limits_{k'=0}^{K'_{\max}-1} \sum\limits_{i=0}^{k'} \left(2K'_{\max} - 2k' - 1\right) u(i, \lambda_{k'}) - \left( \sum\limits_{k'=0}^{K'_{\max}-1} \sum\limits_{i=0}^{k'} u(i, \lambda_{k'}) \right)^2
\tag{7}
$$

### 2.2. Time-Gated SPAD Array Receiver

According to the single time-gated AQ-SPAD photon counting model, we build the AQ-SPADs array photon counting model. We assume that each unit in the SPAD array has the same physical characteristics. For the AQ-SPAD array, blocking time vectors $\mathbf{T_{Bg}}$ can be expressed as:

$$
\mathbf{T_{Bg}} = \{T_{Bg,1}, T_{Bg,2}, \cdots, T_{Bg,n}\}^T.
\tag{8}
$$

We assume that the SPAD in the array is an independent count, and the output of the SPADs array can be expressed as:

$$
Y = \sum_{i=1}^{N_{array}} k_i(\mathbf{T_{Bg,i}}).
\tag{9}
$$

where $k_i$ represents the actual output of each SPAD. The mean and variance of the AQ-SPAD array then can be represented as:

$$
\mu_Y = \sum_{i=1}^{N_{array}} u_{k'_i}(\lambda_{k'}, T_{Bg,i}),
\tag{10}
$$

$$\sigma_Y^2 = \sum_{i=1}^{N_{array}} \sigma_{k'_i}^2(\lambda_{k'}, T_{Bg,i}). \tag{11}$$

According to the mean and variance, the counting model of the time-gated AQ-SPAD array can be approximately calculated as:

$$P(y_n, \mathbf{T_{Bg}}) = \frac{1}{\sqrt{2\pi \sum_{i=1}^{N_{array}} \sigma_{k'_i}^2(\lambda_{k'}, T_{Bg,i})}} \exp\left(-\frac{\left(y_n - \sum_{i=1}^{N_{array}} u_{K'_i}(\lambda_{k'_i}, T_{Bg,i})\right)^2}{2 \sum_{i=1}^{N_{array}} \sigma_{k'_i}^2(\lambda_{k'_i}, T_{Bg,i})}\right) \tag{12}$$

The probability mass function of the time-gated SPAD array is shown in Figure 2. The histogram of the photon counting distribution is fitted with the Gaussian distribution curve. The gate-ON times are defined as $0.25T_C$, $0.5T_C$, $0.75T_C$, and $T_C$. With the increase of gate-ON time, the count means and variance of "1" signal photon and "0" signal photon increase. When the gate-ON time is $0.75T_C$ and $T_C$, the output of the SPAD has ISI distortion. The random blocking time makes the fitting accurate, but basically reflects the photon count statistical characteristics of the time-gated SPAD.

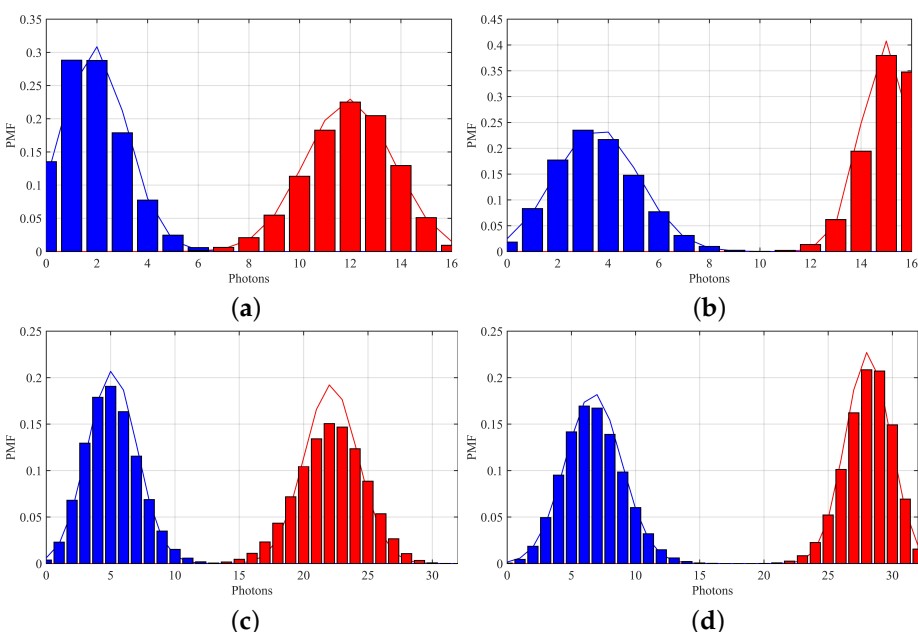

**Figure 2.** The probability mass function of time-gated SPAD array, the distribution of "1" signal and "0" signal is shown in red and blue respectively. ($N_g = 16$, $T_C = 2T_d$, $\lambda_1 = 5/T_C$, $\lambda_0 = 0.5/T_C$). (**a**) $T_g = 0.25T_C$. (**b**) $T_g = 0.5T_C$. (**c**) $T_g = 0.75T_C$. (**d**) $T_g = T_C$.

Figure 3 shows the effect of gate-ON time on the photon detection probability and variance of SPAD. We use photon detection probability and variance as the research objects. The ratio of photon detection probability and variance can be approximately regarded as the signal-to-noise ratio of the time-gated SPAD receiver. When the photon flux is weak, increasing the incident photon can improve the photon detection probability of the SPAD. This is because that low photon flux is the main reason affecting the performance of time-gated AQ-SPAD. Increasing the gate-ON time can effectively improve the photon flux. The detection performance of SPAD increases with the rises of gate-ON time. With the rise of incident photon flux, the photon detection probability of SPAD still benefits from the rise of gate-ON time, and the variance of detection probability tends to be stable. Increasing the gate-ON time is beneficial to the performance of time-gated SPAD. Further increasing photon flux and gate-ON time, the promotion of the SPAD receiver detection

performance is no longer significant. SPAD only requires a shorter gate-ON time to meet the detection performance requirements, the variance is slight, and the statistical characteristics of photons are better. Therefore, when the photon flux is weak, the gate-ON time can be increased to enhance the detection performance of the SPAD. When the photon flux is intense, the time-gated SPAD can be optimized without loss of performance by selecting a reasonable gate-ON time.

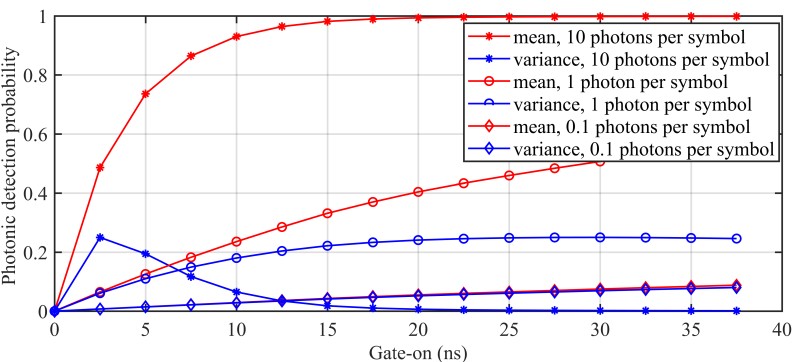

**Figure 3.** The influence of gate-ON time on the photon detection probability and variance.

## 3. Time-Gated SPAD Optimization Technology

### 3.1. BER of Time-Gated SPAD

This section mainly studies the communication performance of time-gated AQ-SPAD. In the transmitting end, ON-OFF keying (OOK) modulation is often considered a reference modulation scheme for evaluating BER performance [24]. In the OOK modulation scheme, the optical signal has only two intensity levels. During each symbol time, the count of the SPAD array is required to determine "0" and "1" signals. The BER of the SPAD receiver consists of the probability that the "0" error is transmitted to "1" and the "1" error is transmitted to "0". If the "0" and "1" signals of the transmitting end have the same probability, the minimum BER can be obtained by intending the intersection of the probability density function of the "0" and "1" signals as the decision threshold. The probability density function of "0" and "1" signals can be expressed as:

$$P_{r1} \sim N\big(u_1(\lambda_1, \mathbf{T_{Bg}}), \sigma_1^2(\lambda_1, \mathbf{T_{Bg}})\big),$$
$$P_{r0} \sim N\big(u_0(\lambda_0, \mathbf{T'_{Bg}}), \sigma_0^2(\lambda_0, \mathbf{T'_{Bg}})\big). \tag{13}$$

Among them, $\mu_1, \sigma_1, \sigma_0, \mu_0$ represent the mean and variance of photon count of "1" signal and "0" signal, respectively. $\mathbf{T_{Bg}}$ and $\mathbf{T'_{Bg}}$ represent the blocking time variables caused by "1" signal and "0" signals, respectively, $\mu_1$ and $\mu_0$ can be calculated by Equation (6), $\sigma_1$ and $\sigma_0$ can be calculated by Equation (7). The photon flux of the "0" signal can be calculated by the background light power, which can be expressed as:

$$\lambda_0 = \frac{RP_b PDE}{N_g hv}, \tag{14}$$

The photon flux of the "1" signal is determined by the background light power and signal light power, which can be calculated:

$$\lambda_1 = \frac{R(P_b + P_r)PDE}{N_g hv}. \tag{15}$$

where $R$ is the responsivity, and the unit is $A/W$, $P_r$ is signal light power, $P_b$ is the background light power, $PDE$ is the detection efficiency, $N_g$ is the number of pixels of the SPAD array, $h$ is the Planck constant, $v$ is the frequency of light signal, and $hv$ usually represents photon energy.

According to the maximum likelihood criterion, the optimal decision threshold for the time-gated SPAD receiver can be calculated as [25]:

$$N_{th} = \frac{\frac{\mu_0}{\sigma_0^2} - \frac{\mu_1}{\sigma_1^2} + \sqrt{\left(\frac{\mu_0}{\sigma_0^2} - \frac{\mu_1}{\sigma_1^2}\right)^2 - \left(\frac{1}{\sigma_0^2} - \frac{1}{\sigma_1^2}\right)\left[\left(\frac{\mu_0^2}{\sigma_0^2} - \frac{\mu_1^2}{\sigma_1^2}\right) + 2\ln\frac{\sigma_0}{\sigma_1}\right]}}{\frac{1}{\sigma_0^2} - \frac{1}{\sigma_1^2}}. \tag{16}$$

Simplify Equation (16) and extract the main terms. The optimal decision threshold can be approximately calculated as:

$$N_{th} = \frac{\mu_1(\lambda_1, \mathbf{T_{Bg}})\sigma_0(\lambda_0, \mathbf{T_{Bg}}) + \mu_0(\lambda_0, \mathbf{T_{Bg}})\sigma_1(\lambda_1, \mathbf{T_{Bg}})}{\sigma_0(\lambda_0, \mathbf{T_{Bg}}) + \sigma_1(\lambda_1, \mathbf{T_{Bg}})}, \tag{17}$$

where, $\mu_1$, $\mu_0$, $\sigma_1^2$, $\sigma_0^2$ represent the mean and variance of the "1" signal and "0" signal, respectively, which is the basis for selecting the appropriate decision threshold. In addition, considering ISI distortion, the decision threshold will be affected by the random vectors $\mathbf{T_{Bg}}$. The optimal decision threshold is usually not an integer and must be determined according to the rounded result.

When transmitting the OOK signal, it is necessary to compare the photon count with the decision threshold $N_{th}$ [3]. When sending the "1" signal, the photon count is compared with the threshold value. If the count value is lower than the decision threshold value, it is considered that the "1" signal transmission is wrong, which can be represented by $P_{e01}$. Similarly, when the "0" signal is transmitted, the photon count is higher than the threshold, it is considered that the transmission is wrong, which can be represented by $P_{e10}$. Based on this, the BER of external time-gated AQ-SPAD can be expressed as:

$$\text{BER} = \frac{1}{2}(P_{e01} + P_{e10}) = \frac{1}{2}\left(\sum_{i=0}^{N_{th}-1} p_1(T_{Bg,i}, \lambda_1) + \sum_{j=N_{th}}^{N_g} p_0(T_{Bg,j}, \lambda_0)\right). \tag{18}$$

where $P_1$ and $P_0$ represent the probability density function of "1" signal and "0" signal, respectively. In actual calculations, it is necessary to perform multiple calculations to obtain average BER to reduce the effects of random vectors $\mathbf{T_{Bg}}$ on the results.

### 3.2. Gate-Width Optimal Technology

At the receiving end, the gate-ON time directly determines the performance of the time-gated SPAD. Appropriate gate-ON time can not only effectively reduce the background photons, but also suppress ISI distortion, to improve the communication performance. However, the signal photon is not detectable in the gate-OFF time, and the gated mode may cause the signal photon counting loss, resulting in a decrease in receiver performance. Therefore, signal photons, background photons, and ISI distortion should be considered in the performance analysis of the time-gated SPAD. Taking the minimization of BER as the optimization target, the blocking time, signal photons, and background photons as constraint conditions, the appropriate gate-ON time is selected to obtain the best BER performance. This optimization problem can be expressed as:

$$\begin{aligned}
&\arg\min_{T_g} BER \\
&\quad K_s = (\lambda_1 + \lambda_b)T_g \\
&\quad K_b = \lambda_b T_g \\
&S.t. \quad 0 \le T_g \le T_s \\
&\quad 0 \le T_{Bg} \le T_g
\end{aligned} \tag{19}$$

In order to solve the optimization problem, according to Equations (6) and (7), the random variable blocking time vectors $\mathbf{T_{Bg}}$ also affects the optimal gated-ON time in addition to $K_S$ and $K_b$. ISI distortion can only be eliminated when the gate-OFF time exceeds the blocking time. In addition, the concavity and convexity of the BER function are affected by

$K_S$ and $K_b$. In this paper, the optimal gate-ON time is obtained by numerical simulation, and the accuracy of the results is verified.

## 4. Simulation and Discussion

In this section, the optimal gate-ON time is obtained by Monte Carlo numerical simulation. In this simulation, the PDE is set to 0.45, the dead time is 25 ns, and the symbol time is 37.5 ns. Two sizes of SPAD receivers are considered, including $N_g = 16$ small-scale array and $N_g = 64$ medium-scale array. For $N_g = 16$ SPAD array receivers, the decision threshold has a great impact on the BER. The array with $N_g = 64$ can reduce the impact of the decision threshold on the change of BER. The experimental results have a SPAD array of $N_g = 16$, which retains the influence of the decision threshold on the BER. For the SPAD array with $N_g = 64$, the effect of decision threshold on BER fluctuations is lowered, and the BER curve is smoother. The parameters of the SPAD system are referred to Refs. [2,26] and are shown in Table 1.

**Table 1.** SPAD system parameter table.

| Characteristics Parameter | Value |
|---|---|
| Photon Detection Efficiency @550 nm | 0.45 |
| Dead time | 25 ns |
| Symbol time | 37.5 ns |
| Gate-On time | 0~37.5 ns |
| Number of SPADs | 16/64 |

### 4.1. Impact of Gate-ON Time on Receiver Performance

According to the BER performance of SPAD, $K_S$, $K_b$, and $T_g$ are the main factors affecting the time-gated SPAD receiver. Firstly, the influence of different gate-ON time on BER is analyzed.

The performance of time-gated SPAD is observed by setting different gate-ON times. To explain the results, we set different signal photon flux and background photon flux to verification. The experimental results are shown in Figure 4. When the background photon flux or signal photon flux are weak, increasing the gate-ON time can effectively improve BER performance. Further increasing the background photon flux, the SPAD receiver has an optimal gate-ON time.

In order to further explore the functional relationship between gate-ON time, background photon, and signal photon, we set the target BER as $10^{-3}$ and fixed the gate-ON time as 10 ns, 15 ns, and 20 ns. By changing the background photon, we observe the number of minimum signal photons to achieve the target BER. The results are shown in Figure 5. When the background photon is weak, $T_g = 20$ ns requires the least signal photons to achieve the target BER. Further increasing the background photon flux, the longer the gate-ON time, the more signal photon flux required to achieve the target BER. The reason for this result is that as the background photon increases, the BER performance of the receiver is mainly affected by the background photon, and the increasing gate time may result in more background photon incidents, making the BER performance decrease. If a shorter gate-ON time is selected, the influence of background photons on BER can be reduced, and fewer signal photons are to achieve the target BER. The SPAD array receiver of $N_g = 64$ is selected, and the experimental results are shown in Figure 5. When the background light flux is weak, the gate-ON time $T_g = 20$ ns requires the least signal photons to achieve the target BER, but different gate-ON times are less affected by BER performance. With the increase of background photon flux, when the gate-ON time is 20 ns, more signal photons are needed to achieve the target BER, and when the gate-ON time is 10 ns, fewer signal photons are required to achieve the target BER. Therefore, the SPAD's optimal gate-ON time needs to be selected according to the background photons and signal photons.

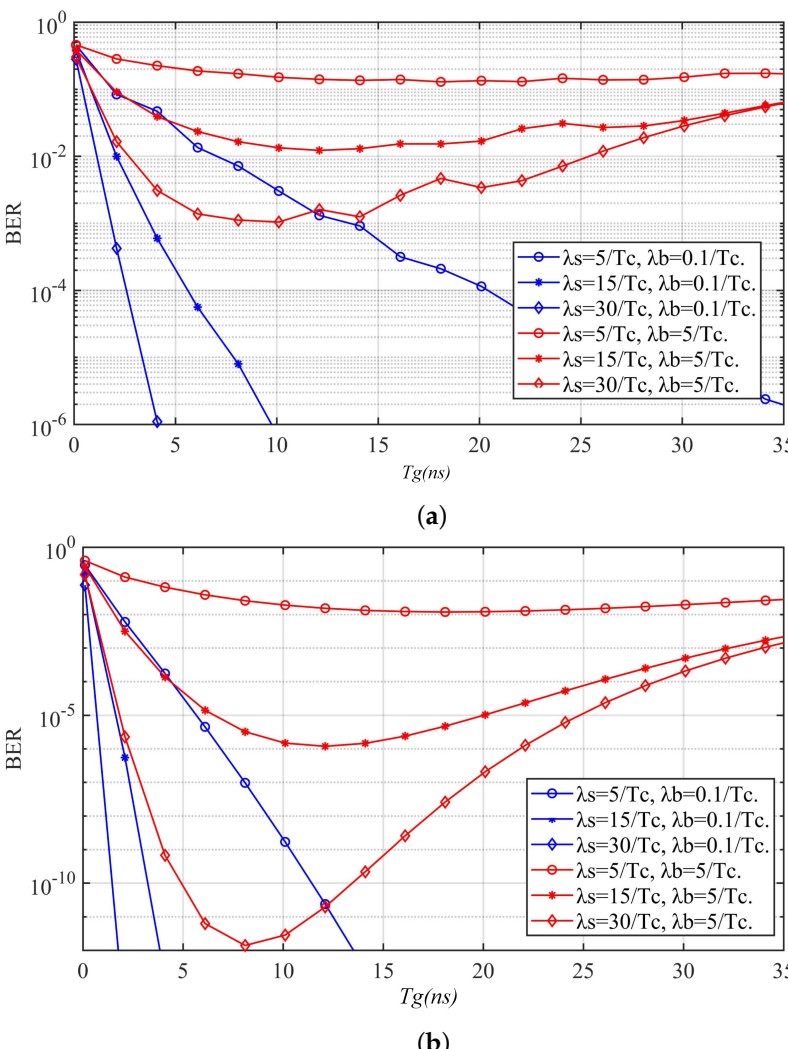

**Figure 4.** The influence of gated-ON time on the performance of time-gated SPAD. (**a**) $N_g$ = 16. (**b**) $N_g$ = 64.

*4.2. Optimal Gate-ON Time*

In order to obtain the optimal gate-ON time, the background photon flux of different intensities is selected. Signal photons are independent variables, and the optimal gate-ON time is obtained by numerical simulation. The results are shown in Figure 6. When the signal photon flux and background photon flux are weak, the gate-ON time is the main factor affecting the performance of the SPAD receiver, and the shorter gate-ON time will affect the photons incidence, the optimal gate-ON time is the symbol time $T_C$.

The output photon count of SPAD has an exponential relationship with the signal photon. Further increasing the signal photon flux, the SPAD output does not exceed the maximum count. Although the gated mode reduces the incidence of signal photons, it has little effect on the actual counting output, and the appropriate gate-ON time can suppress the incidence of background photons and improve the BER performance.

When the signal light flux is low and the background light flux is high, the lower bound of BER is affected by the background light. Appropriate gate-ON time needs to be set to reduce the influence of background photons on communication performance and improve the performance of SPAD. When the signal light flux and the background light flux are high, the appropriate gate-ON time can effectively reduce the signal photons and the background photons, so that the signal photons and the background photons reach the appropriate ratio to enhance the performance of the SPAD. The results of Figure 6 can be explained reasonably by the above situation.

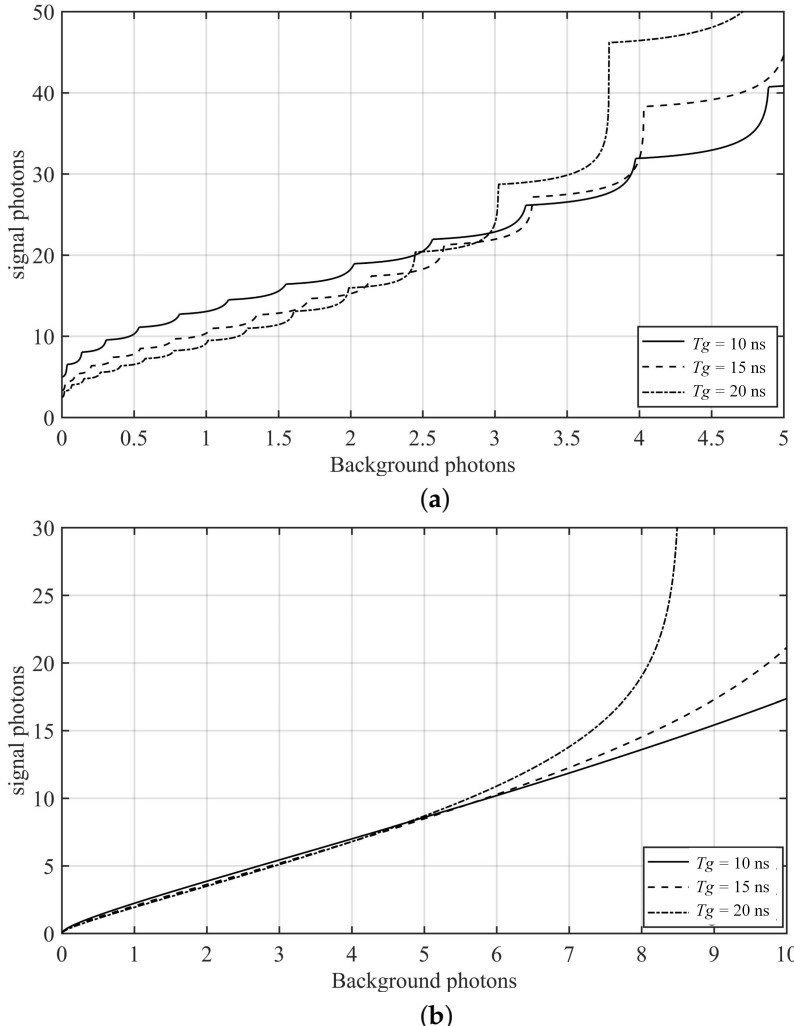

**Figure 5.** Background photons and signal photons required to achieve the target BER. (**a**) $N_g$ = 16. (**b**) $N_g$ = 64.

When the signal and background photon is low, the optimal external gate-ON time is symbol time. As the signal photon flux increases, the optimal gate-ON time is less. On the one hand, the SPAD has the upper limit of the photon count. Excessive signal photon incident does not change the maximum count, and the gated mode reduces the signal photon incident, but the signal photon flux is sufficient, and the BER performance is less affected; on the other hand, the gated mode reduces the incidence of the background photon, alleviates the counting error of the SPAD and enhances the BER performance. In other words, the gated mode reduces the signal photon flux, resulting in the loss of BER performance, but it also reduces the background photon flux and improves the BER performance. By weighing the performance loss and performance gain caused by the gated mode, the optimal gate-ON time is obtained to achieve the optimal performance of time-gated AQ-SPAD. When the background photons flux is high, the lower bound of the BER is determined by the background photons. The gated mode reduces the influence of the background light flux, improving the communication performance of the SPAD.

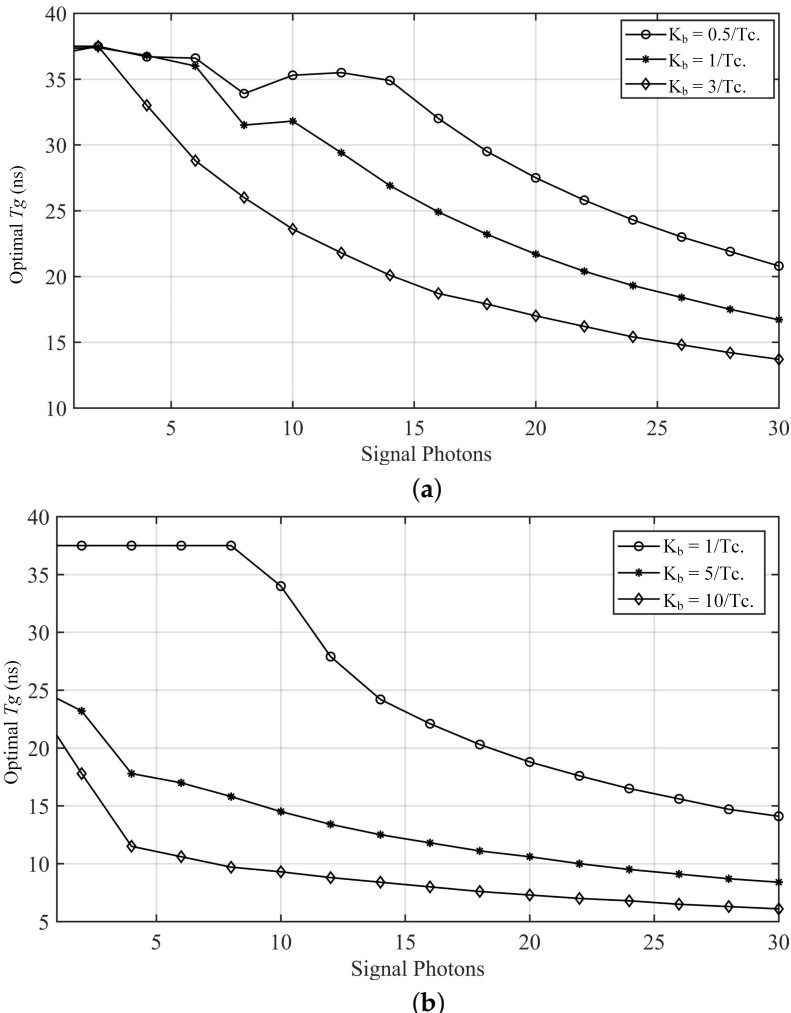

**Figure 6.** The optimal gate-ON time under different signal intensity. (**a**) $N_g$ = 16. (**b**) $N_g$ = 64.

### 4.3. The Performance Comparison of the Gated Mode and the Free-Running Mode

In order to further illustrate the results of Figure 6, the BER performance of gated mode and free-running mode is compared, and the results are shown in Figure 7. The gated mode is operated in the optimal gate-ON time, and the free-running mode does not have an additional external gate signal, the SPAD can be exposed in the whole symbol cycle. When the signal flux is low, the performance of a free-running SPAD is equivalent to the time-gated SPAD. With the increase of signal photon flux, the BER of the free-running SPAD receiver decreases and reaches the lower limit. The lowest BER of free-running AQ-SPAD is determined by the background light. For time-gated SPAD, the performance is equivalent to free-running mode at low signal light flux. As the signal increases, the BER of free-running SPAD achieves the lower limit. The BER of the time-gated SPAD decreases monotonously, the background light has a small impact, and the time-gated SPAD can achieve a lower BER.

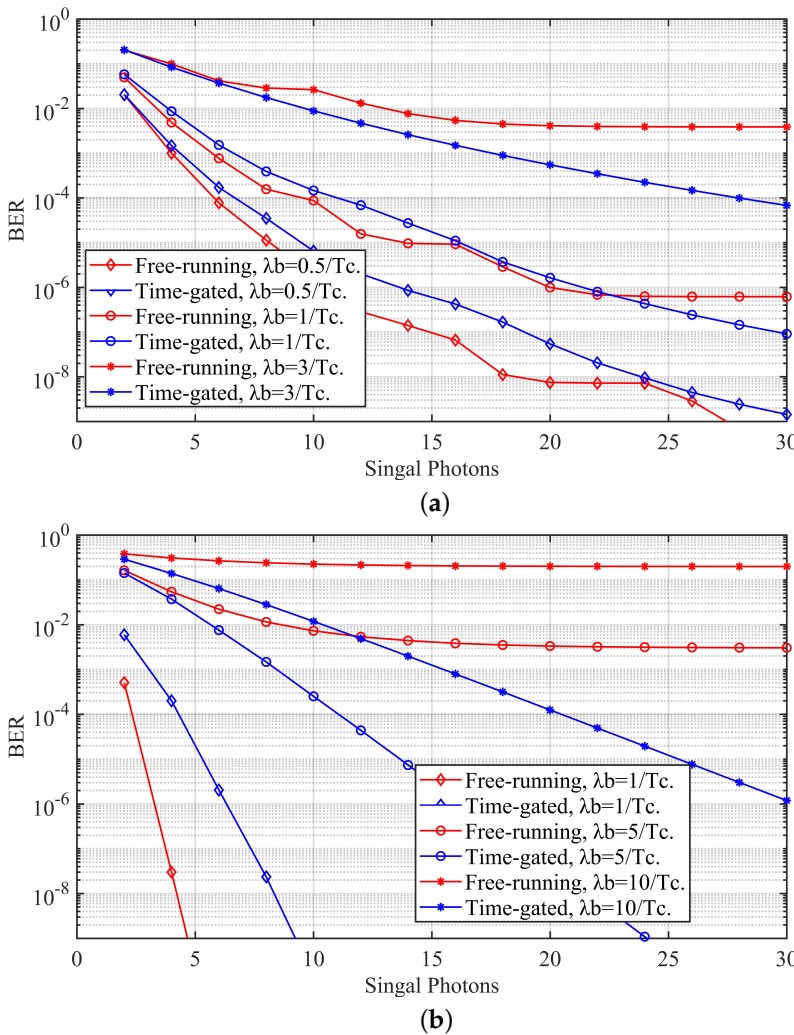

**Figure 7.** Performance comparison of time-gated SPAD and free-running SPAD. (**a**) $N_g$ = 16. (**b**) $N_g$ = 64.

## 5. Conclusions

In this paper, we use the time-gated AQ-SPAD as a receiver to achieve better communication performance by optimizing the gate-ON time. The photon counting model based on external time-gated AQ-SPAD is established according to counting characteristics and ISI distortion. The BER model of the time-gated AQ-SPAD receiver is constructed. In addition, we use the minimum BER as an optimized target, and determine the optimal gate-ON time according to the signal photon flux and background photon flux. Numerical experiments show that the gated mode enhances the background of the AQ-SPAD receiver. Moreover, the gated mode has better BER performance than the free-running mode, especially when the signal light flux or the background light flux is higher, and the BER performance of the time-gated mode is apparently better than the free-running mode.

**Author Contributions:** Conceptualization, Y.M. and Y.Z.; methodology, Y.M.; validation, Y.M. and X.D.; formal analysis, Y.M.; writing—original draft preparation,Y.M.; writing—review and editing, Y.Z., C.W., Z.Y. and X.D. All authors have read and agreed to the published version of the manuscript.

**Funding:** The research of this paper was funded in part by the National Key Research and Development Project under Grant No. 2018YFB1801903, in part by the National Natural Science Foundation of China (NSFC) under Grant 61901524 and in part by the China Postdoctoral Science Foundation under Grant No. 2019M663477.

**Institutional Review Board Statement:** Not applicable.

**Informed Consent Statement:** Not applicable.

**Data Availability Statement:** Not applicable.

**Acknowledgments:** The authors wish to thank the anonymous reviewers for their valuable suggestions.

**Conflicts of Interest:** The authors declare no conflict of interest.

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
