# Peer review of "Gate-Width Optimisation Based on Time-Gated Single Photon Avalanche Diode Receiver for Optical Wireless Communications"

_electronics, doi:10.3390/electronics11142218_

Round 1

Reviewer 1 Report

Please check the minor spelling and citations of some basic Equations for final version.

Reviewer 2 Report

Dear Authors,

In your paper 'Gate-Width Optimal based on Time-Gated Single Photon Avalanche Diode Receiver for Optical Wireless Communications' you analyzed the performance of wireless communication based on the gate-time of a single photon counting receiver. From my point of view it is adequate to determine the bit error rate with the help of a Monte-Carlo simulation with random numbers of 'blocking times' (T_B_g) but since your input has all dependencies available and for the random numbers of T_B_g a statistical distribution has to be assumed anyway I am wondering why it is not possible to calculate the bit error probability analytically. Additionally, for the analysis a perfect jitter-free synchronization of the sender and receiver is assumed. How this synchronization should be made in order to exactly trigger the gating of the single photon counting device is not clear. Despite these general remarks there are a couple of errors that must be corrected in order to make the paper publishable:

1) You should check that all used mathematical symbols are consistent explained before use:

1a) In Fig. 1 the gate-ON time is T_ON, T_g is never defined but one might guess that it is the same as T_ON? 

1b) What is the difference between T_B and T_B_g?

1c) What is k' in (4)?

1d) Is K'_MAX in (3) the same as k'_MAX in the following?

1e) What is N_ARRAY in (9)? I guess it is n as used in (8).

1f) What is k_i in (9)?

1g) How is u_K_i in (10) defined? I guess it should read µ_K_i?

1h) T_B_g,i in (10) and (11) should not be bold, these are the elements of a vector but not a vector. Should I assume lambda'_K in (10) and (11) is lambda_K'? 

1i) I assume that lambda_K'_i in (12) should be same as lambda_k'_i? Moreovdr, there seems an i index is missing in the normalization factor at lambda_K'.

2) If variance and mean is defined by (10) and (11), why they are not used in (12)?

3) I assume 12 is a normal distribution. Than a square and a minus sign is missing in the exponent or should it be an exponential decay, but than the normalization factor does not make sense.

1j) What is the meaning of N(..) in (13)? Is it again the normal distribution? If yes why this notation is not used in (12)?

1k) What is u_1 and u_0 in (13)? You mean µ_1 and µ_0?

4) Fig. 3 is completely unclear. The gate_ON time is given in ns, the first time ever an absolute value is used and so far completely unrelated to all the other time values. To relate it to anything one has to read until the end, where absolute timing is used. This make the paper hard to understand. Moreover it is not clear what the different curves mean. The description in the text is not sufficient. The meaning of all curves should be described and what the result is.

5) What is the . in (14) and (15)? Multiplication?

6) What is the background that the photon flux is proportional to 1/N_g? Photon flux is photon flux and independent of measurement. The measurement is count rate. Why PDE and R are the same if N_g is changing? What is the background of this assumption?

7) In fig. 2 the Gaussian fit do not really fit, especially not in c).

8) The meaning of the vector T_B_g remain unclear until reading to line 197, what makes it clear that an array of random numbers are meant. This should be made clear before at its definition.

9) In (18) P_e01, P_e10, P_1 and p_0 are undefined or should I read [22] in order to understand your paper?

10) Since lambda_l and lambda_b defined as rates, K_s and K_b are count numbers having no unit. But in fig. 4 Ks and Kb what I assume are the same as K_s and K_b have the unit 1/s since T_C is a time. This does not fit together.  

I think with a careful revision the paper could be made acceptable. Nevertheless I suggest that it would be much better if absolute times are not used at all and all times are simply normalized with the help of T_C, that make everything much more clear.

Reviewer 3 Report

The overall manuscript looks useful.

Some language issues such as missing articles or spaces (marked blue in screenshot) and grammar issues (marked red in the first two  pages) require typesetting changes. Pages 3-12 will still need proofreading.

There are few technical points that are unclear and need to be addressed as follows:

abstract, line 10: the optimal gate-ON time is not ‘given’ (as input) but ’derived’ (as output)

Section 1, line 29: what is meant by the phrase ‘deal some effects to’?

Section 1, lines 39-40 and 46-47:  the argument regarding the fill factor is correct only if one compares pixels of same total area. In principle it is not clear per se whether 1 pixel with active area A is worse or better than N active pixels each with active area A/N; the authors may wish to comment on aspects of shot noise and background signal height influencing that decision.

Section 1, line 52: some noun seems to be missing after the adjective ’biomedial’, e.g. ‘devices’ or ‘sensors’ or ‘applications’ - please choose and elaborate.

Section 4, line 205 ff: Firstly, this work does not seem to present any ‘experiment’ at all but only deals with simulations instead. There should either be an explanation or a reference to previously published work for the specific set of parameters chosen for the simulations here, as given at the bottom of page 7.

Please explain where this manuscript goes beyond that cited as ref. 2

Round 2

Reviewer 2 Report

Dear Authors,

As far as I can see all corrections are sufficient and I have now voted for publishing as it is. Thank for for considering my suggestions.

This manuscript is a resubmission of an earlier submission. The following is a list of the peer review reports and author responses from that submission.

Round 1

Reviewer 1 Report

The paper sentences and grammar need to be improved.

The output results required justification by comparing with other researcher work.

The experimental work is by simulation only or fabrication? Because the output graph plotted  is in consistent. Would be good to have a figure of the fabricated hardware.

Reviewer 2 Report

This paper is interesting.

It should be accepted.

Reviewer 3 Report

A time-gated SPAD optimization method is studied for the OOK modulation - based optical wireless communication system.

BER performance of the time-gated mode is evaluated and compared to free-running mode.

Consider the following comments for improving the manuscript:

+ Rewrite the Introduction part with adding more recent references. Present clearly contributions of the paper?

+ It is better to provide the Table of parameters and constants in the part 4 (experiment and discussion)

+ Check typos and gramma through the paper, for example, the decision threshold N_th N_th (page 6), N_g (page &), and so on.

Reviewer 4 Report

This paper discusses the effect of time-gated single photon avalanche diode receiver and investigate the optimized gate time. The theme is appropriate for Electronics, but the reviewer has to recommend for rejection. Some of the main reasons are that authors’ proposed model is unclear, there are undefined variables to clarify the models and the characterizations, and the presentations of the calculated results are poor. The reviewer would like to ask authors to carefully sophisticate the presentations for the proposed model and their estimated characteristics. After that, authors can explain the scientific novelty and emphasize the strong points of the proposed model toward the advanced optical wireless communication based on quantum information.

Reviewer 5 Report

10,11 line- It could be explained what numerical results or how much better. 

How can your calculations be applied in practice?

The article is written clearly, all calculations and results are presented.